# Don’t Forget the Bones: Incidence and Risk Factors of Metabolic Bone Disease in a Cohort of Preterm Infants

**DOI:** 10.3390/ijms231810666

**Published:** 2022-09-14

**Authors:** Michela Perrone, Amanda Casirati, Stefano Stagi, Orsola Amato, Pasqua Piemontese, Nadia Liotto, Anna Orsi, Camilla Menis, Nicola Pesenti, Chiara Tabasso, Paola Roggero, Fabio Mosca

**Affiliations:** 1Neonatal Intensive Care Unit, Fondazione IRCCS Ca’ Granda Ospedale Maggiore Policlinico, 20122 Milan, Italy; 2Clinical Nutrition and Dietetics Unit, Fondazione IRCCS Ca’Granda Policlinico San Matteo, 27100 Pavia, Italy; 3Department of Health Sciences, University of Florence, Anna Meyer Children’s University Hospital, 50139 Florence, Italy; 4Department of Clinical Sciences and Community Health, University of Milan, Via Francesco Sforza 35, 20122 Milan, Italy; 5Department of Statistics and Quantitative Methods, Division of Biostatistics, Epidemiology, and Public Health, University of Milano-Bicocca, 20126 Milan, Italy

**Keywords:** prematurity, bone disease, nutrition

## Abstract

Metabolic bone disease of prematurity (MBD) is a condition of reduced bone mineral content (BMC) compared to that expected for gestational age (GA). Preterm birth interrupts the physiological process of calcium (Ca) and phosphorus (P) deposition that occurs mostly in the third trimester of pregnancy, leading to an inadequate bone mineralization during intrauterine life (IUL). After birth, an insufficient intake of Ca and P carries on this alteration, resulting in overt disease. If MBD is often a self-limited condition, in some cases it could hesitate the permanent alteration of bone structures with growth faltering and failure to wean off mechanical ventilation due to excessive chest wall compliance. Despite advances in neonatal intensive care, MBD is still frequent in preterm infants, with an incidence of 16–23% in very-low-birth-weight (VLBW, birth weight <1500 g) and 40–60% in extremely low-birth-weight (ELBW, birth weight <1000 g) infants. Several risk factors are associated with MBD (e.g., malabsorption syndrome, parenteral nutrition (PN), pulmonary bronchodysplasia (BPD), necrotizing enterocolitis (NEC), and some chronic medications). The aim of this study was to evaluate the rate of MBD in a cohort of VLBWI and the role of some risk factors. We enrolled 238 VLBWIs (107 male). 52 subjects were classified as increased risk (G1) and 186 as standard risk (G2) according to serum alkaline phosphatase (ALP) and phosphorus (P) levels. G1 subjects have lower GA (*p* < 0.01) and BW (*p* < 0.001). Moreover, they need longer PN support (*p* < 0.05) and invasive ventilation (*p* < 0.01). G1 presented a higher rate of BPD (*p* = 0.026). At linear regression analysis, BW and PN resulted as independent predictor of increased risk (*p* = 0.001, *p* = 0.040, respectively). Preventive strategies are fundamental to prevent chronic alteration in bone structures and to reduce the risk of short stature. Screening for MBD based on serum ALP could be helpful in clinical practice to identify subjects at increased risk.

## 1. Introduction

Osteopenia or metabolic bone disease of prematurity (MBD) identify a condition of reduced bone mineral content compared to that expected for gestational age, associated with alterations in biochemical and radiological indices [1,2]. MBD is a multifactorial disease, mainly caused by inadequate bone mineralization resulting from various conditions that can develop both during intrauterine life and after birth [3]. It is well-known that prematurity affects the bone mineralization process; indeed, preterm birth interrupts the physiological process of calcium (Ca) and phosphorus (P) storage and deposition during the period of maximum growth, i.e., the third trimester of pregnancy. Indeed, during the last three months of pregnancy, calcium and phosphorus deposition occurs at a rate of 120 mg/kg/day and 60 mg/kg/day, respectively [4].

After preterm birth, an inadequate intake of calcium and phosphorus carries on these alterations, resulting in the development of overt disease [5].

The diagnosis of MBD is mainly biochemical through specific indices of bone turn-over. It includes evaluation of serum levels of calcium, phosphorus, parathyroid hormone (PTH) and alkaline phosphatase (ALP) and urinary excretion of calcium and phosphorus. The main change in biochemical indices associated with MBD is the reduction of phosphoremia to concentrations below 4.5 mg/dl [6]. Hypophosphatemia is an early indication of calcium/phosphorus metabolism alteration with a sensitivity of 100% and specificity of 94% in the diagnosis of MBD [6], while ALP levels > 900 IU/L have 100% sensitivity and 70% specificity in MDB diagnosis [7]. Previous studies have demonstrated that ALP levels above 700 IU/L and phosphorus levels below 4.5 mg/dl at three to four weeks of life are related to increased risk of developing MBD in preterm infants [8,9]. Instrumental evaluation of bone health status is difficult in the neonatal period. Faienza and coworkers discussed the different instrumental exams available to assess bone health (i.e., X-Ray, dual-energy X-ray absorptiometry and quantitative ultrasound [10]. X-rays can be performed in newborns, even if they are in unstable conditions. However, they do not identify early changes in bone mass; therefore, they are not indicated as a screening test [11,12]. Regarding dual-energy X-ray absorptiometry, it is considered as gold standard for the assessment of bone mineral density [13]. However, it is difficult to apply this method in the neonatal period because it requires moving the infant out of the neonatal intensive care department. Moreover, it is influenced by artifact movements. Liao and coworkers have identified quantitative ultrasound as a noninvasive, radiation-free and portable method to evaluate bone health status in newborns [14]. Soto Martinez, in a study performed on deceased newborns and infants, demonstrated that quantitative ultrasound reflects the bone mineral contents [15]. Nevertheless, this method requires trained clinical staff.

The American Academy of Pediatrics Committee on Nutrition (AAP-CON) recommends to screen all very-low-birth-weight infants through the measurement of serum levels of phosphorus and alkaline phosphatase around the fourth week of life [16].

Despite advances in neonatal intensive care and implementation of nutritional support, MBD is still frequent in preterm infants, with an incidence of 16–23% in very-low-birth-weight (VLBW, birth weight <1500 g) infants and in 40–60% in extremely low-birth-weight (ELBW, birth weight <1000 g) infants [11,17]. Moreover, MBD occurrence is inversely related to gestational age and birth weight [3,18], with a peak of incidence at 3–5 weeks of life [19].

Several risk factors are associated with the development of MBD, including malabsorption syndrome, prolonged immobility, total parenteral nutrition (PN) for more than 4 weeks, pulmonary bronchodysplasia (BPD), necrotizing enterocolitis (NEC), liver disease, chronic kidney disease and medication (as chronic use of diuretics) [1,2,6,20,21].

Moreover, even if in the majority of cases MBD is a self-limiting condition, in some cases it could lead to a permanent alteration of bone structure that could cause growth faltering and alteration of respiratory patterns due to excessive chest wall compliance [19].

Considering the increased risk of preterm infants to develop MBD and the possible consequences in later life, the present study aimed to evaluate the rate of MBD in a cohort of VLBW infants and the correlation with clinical data and known risk factors.

## 2. Results

A total of 289 VLBWIs were enrolled in the present study; 51 infants died during the first six weeks of life (mean days of life at death was 10.27 ± 7.5 days, range 1–31), so 238 subject with a mean gestational age of 29.2 ± 1.98 weeks and mean birth weight of 1131.13 ± 258.20 g were included in the final analysis (male 45%, twins 51%). According to serum P and ALP concentration performed at 25.84 ± 3.45 days of life [19,20,21,22,23,24,25,26], subjects were divided into two groups: G1 (=52 subjects) infants were at increased risk of MBD as they had P ≤ 4.5 mg/dl or ALP ≥ 900 UI/L, while G2 (=186 subjects) infants did not have an increased risk of MBD. Table 1 reports clinical data in G1 and G2. Table 1 reports basic characteristics.

As shown in Table 1, G1 subjects had lower gestational age (28.53 ± 1.99 vs. 29.38 ± 1.94 weeks, *p* < 0.01) and birth weight (1019.87 g ± 243.57 vs. 1162.23 g ± 245.16, *p* < 0.001). Moreover, they needed a longer parenteral nutrition support (25.35 days ± 10.51 vs. 20.54 days ± 13.55, *p* < 0.05) and invasive ventilation (7.46 ± 9.27 days vs. 3.41 ± 10.05 days, *p* < 0.01).

To better understand the influences of the main comorbidities, we analyzed the rate of necrotizing enterocolitis, cholestasis, pulmonary bronchodysplasia, intraventricular hemorrhage, retinopathy of prematurity and sepsis into groups. A χ2-test was applied in order to evaluate differences in G1 vs. G2. Fisher’s test was applied instead of the χ2-test if the numerosity of a single cell was <5. G1 subjects presented a higher rate of BPD (36.5% vs. 20.4%, *p* = 0.026).

Although there was a tendency towards a higher incidence of sepsis in the group at risk of MBD, this difference did not reach statistical significance (Table 2).

At multivariate linear regression analysis, birth weight and parenteral nutrition duration resulted as independent predictors of increased risk of MBD (Table 3)

Five patients (2%) presented radiological signs of osteopenia at X-ray performed at discharge. Specifically, four patients presented a reduction in bone mineral content and one patient presented an increased submetaphyseal lucency. No cases of spontaneous fractures occurred in our population. These patients were of a birth weight of less than 1000 g and had ALP > 1000 UI/L. Moreover, they experienced prolonged mechanical ventilation due to severe pulmonary bronchodysplasia. X-ray pictures relative to the five patients with radiological signs of osteopenia are shown in the Appendix A.

Twenty-two subjects were discharged with oral therapy with fructose 1–6 bisphosphate and calcium gluconate. They needed to continue oral supplementation until normalization of serum ALP levels, which occurs at a mean age of 46.85 ± 0.95 post-conceptional weeks.

## 3. Discussion

Preterm infants have an increased risk of developing abnormalities in bone tissue due to reduced mineral reserves. Although in most cases it is a self-limiting phenomenon, sometimes it could lead to permanent changes in bone structure [22].

In this study we investigated an increased risk of developing MBD and the presence of known risk factors in a cohort of very-low-birth-weight infants.

The rate of MBD found in our population reflects what is reported in the literature, i.e., 22% VLBWI with gestational age < 32 weeks are at an increased risk of developing MBD during hospital stays [10,20,21,22,23].

This is probably a consequence of the fact that the highest bone mineralization occurs in the last trimester of pregnancy, and that intrauterine growth retardation may be associated with placental dysfunction, and therefore a reduction in nutrient supply from mother to fetus [2,25]. As previously reported, low birth weight is one of the main factors associated with the risk of MBD. Indeed, while no patients with birth weight >1000 g developed the overt disease, five patients (5.8%) with birth weight < 1000 g presented signs of metabolic bone disease at discharge, confirmed by the presence of specific alteration in bone mineralization or structure at X-ray analysis.

These data confirm data reported in a retrospective study published by Viswanathan et al. in 2014. The author compares 71 premature infants who developed MBD with 159 healthy controls. The MBD group has a lower gestational age and birth weight than the healthy controls [17]. Ukarapong et al. also find the same association between birth weight and increased risk of MBD. In addition, a positive correlation between MBD and duration of parenteral nutrition is demonstrated in the same study, as well as in our case studies, while our data do not confirm what they reported about the increased incidence of cholestasis in the group at increased risk of MBD [2,26].

The increased length of the stay in our population at risk confirms what was reported in previous study by Chen and coworkers [25,27].

A significant difference was also found in the incidence of bronchodysplasia and in the longer duration of mechanical and noninvasive ventilation in infants at increased risk. These data confirm the findings of the previous published studies [2,10,27,28].

A higher incidence of bronchodysplasia and the prolonged duration of artificial ventilation were also found in a study published by Avila-Alvarez and colleagues, although the differences in this case do not reach statistical significance [29]. Probably, this increased rate of MBD in subjects affected by pulmonary bronchodysplasia is secondary to the chronic use of such drugs as diuretics, methylxanthines and corticosteroids [30]. A negative influence on bone mineralization is played by prolonged immobility due to the need for mechanical ventilation for long periods. This aspect was investigated by Torrò-Ferrero and coworkers in a randomized controlled trial using tibial speed of sound (tibial-SOS) as marker of bones ‘health. It results in subjects who received reflex locomotion therapy instead other physiotherapy procedures presenting an improvement in tibial-SOS levels at the end of intervention [31]. Nevertheless, as pointed out by Kavurt and colleagues, it is difficult to understand if metabolic bone disease is a cause or consequence of pulmonary bronchodysplasia. Indeed, if the chronic use of some medications and prolonged immobility due to mechanical ventilation could be the first step of alteration in the deposition of mineral content in bones, on the other hand the alteration in bone mineralization and structure could itself influence chest wall compliance and make it difficult to wean the patients from ventilator support [19].

The strength of the present study is that we included a relatively large cohort of VLBWIs and all subjects were managed (nutritionally and medically) in a standardized manner according to internal procedures. The limit is that we conducted a retrospective observational study.

## 4. Materials and Methods

### 4.1. Study Design

A retrospective study was conducted. Preterm infants were enrolled at birth and evaluated at 25.84 ± 3.45 days of life.

### 4.2. Subjects

Very-low-birth-weight infants (birth weight < 1500 gr) with gestational age ≤ 32 weeks, born between January 2017 and December 2020 and admitted to the Neonatal Intensive Care Unit at Fondazione IRCCS Ca’ Granda Ospedale Maggiore Policlinico in Milan, were enrolled. Subjects who died during the first six weeks of life and with genetic abnormalities were excluded from analysis. According to AAP-CON guidelines [16], serum *p* and ALP concentration were dosed between 3rd and 5th week of life. Subjects were classified as “increased risk” (G1) if P ≤ 4.5 mg/dl or ALP ≥ 900 UI/L, or “standard risk” (G2) if P ≥ 4.5 mg/dl and ALP ≤ 900 UI/L [7].

### 4.3. Clinical Data Collection

Infants’ baseline characteristics, days of PN, days on mechanical ventilation, length of hospital stay and incidence of comorbidities such as necrotizing enterocolitis (stage II-b to III-b according to Bell’s criteria [32]), pulmonary bronchodysplasia (need for chronic oxygen supplementation at 36 weeks of correct age), cholestasis (conjugated bilirubin >2 mg/dl), sepsis (defined as the presence of clinical signs associated with increased inflammatory index and positive blood culture), retinopathy of prematurity and intraventricular hemorrhage grade III or more were collected from patients’ computerized clinical charts. Serum P and ALP concentrations were also collected between the 3rd and the 5th weeks of life, during routine blood draws. Serum P was measured using a direct phosphomolybdate reaction (PHOS2, Cobas c Roche/Hitachi), while ALP activity was determined colorimetrically by p-nitrophenyl phosphorus method (ALP2L, Cobas c Roche/Hitachi).

### 4.4. Nutritional Practice

During hospitalization, infants were fed according to internal procedures for nutrition in VLBW infants [33]. Parenteral nutrition (PN) was started on the first day of life using personalized bag according to ESPGHAN guidelines, and the infused volume was gradually increased during the first week of life to achieve a volume of 150–180 mL/kg at the end of the first week of life. According to ESPGHAN recommendations, PN provided 60–65 kcal/kg/day with 3.5–4.5 g/kg/day of protein. PN infusion was progressively decreased until suspension as infants assumed an enteral volume of 50 mL/kg/day [34,35].

Regarding enteral feeding, it was started as soon as possible during the first days of life. Fresh own mother’s milk was the first choice; if milk from the infant’s mother was not available or insufficient, pasteurized donor human milk was chosen. When an enteral volume of 80 mL/kg/day was reached and enteral nutrition was adequately tolerated, a target fortification using multicomponent bovine-based fortifiers was started in order to achieve an enteral energy intake between 110 and 160 kcal/kg/day, with a protein intake of 3.5–4.5 g/kg/day, as recommended in ESPGHAN guidelines [35,36].

Moreover, all subjects were supplemented with fructose 1–6 bisphosphate and calcium gluconate according to internal procedure [33].

### 4.5. Anthropometric

Weight, length and head circumference were measured by trained paramedical staff from our institution according to standard procedure.

Weight was measured using the incubator integrated scale, while length was measured to the nearest 1 mm on a Harpenden infantometer, which has a fixed headboard and moveable footboard. Head circumference was measured to the nearest 1 mm using non-stretch measuring tape [37].

Weight, length and head circumference z-score were calculated using the z-score calculator provided by Intergrowth-21st project [31].

### 4.6. Radiological Examination

Chest C-rays were performed before discharge to evaluate the presence of radiological signs of metabolic bone disease. According to Koo’s criteria, a pediatric radiologist reviewed the X-ray exams in order to find the following signs of alteration in bone mineralization and/or structure: presence of loss of dense zone of provisional calcification at metaphysis, increased submetaphyseal lucency, thinning of cortex, fraying, splaying or cupping of metaphysis [38].

### 4.7. Statistical Analysis

Continuous variables are presented as mean and standard deviation, while categorical variables are expressed as absolute numbers or percentages. Differences between G1–G2 were evaluated using t-test for continuous variables and χ2-test for categorical variables or Fisher’s test if cell had fewer than 5 elements. All statistical analysis was performed by SPSS 20 statistical package. A multivariate linear regression analysis was applied to evaluate the influences of known risk factors (as birth weight, gestational age, length of parenteral nutrition, length of mechanical ventilation). The statistical significance level was fixed at 0.05.

## 5. Conclusions

Despite nutritional approach and early supplementation with calcium, phosphorus and vitamin D, preterm infants have an increased risk of MBD, especially if born with a weight below 1000 g. Screening for MBD based on serum ALP could be helpful in clinical practice to identify subjects at increased risk. Further studies are needed to explore the pathophysiological mechanisms underlying the development of MBD, in order to implement preventive therapies to reduce the incidence of this complication in a susceptible population that is still elevated.

## Figures and Tables

**Table 1 ijms-23-10666-t001:** Clinical data. GA, gestational age; BW, birth weight; PN, parenteral nutrition; LOS, length of stay; NIV, noninvasive ventilation; ALP, alkaline phosphatase; P, phosphorus. G1, subjects at increased risk of metabolic bone disease. G2, subjects with standard risk of metabolic bone disease.

	G1 (52)	G2 (186)	
	Mean ± SD/N(%)	Mean ± SD/N(%)	*p*
GA (weeks)	28.53 ± 1.99	29.38 ± 1.94	<0.01
BW	1019.87 ± 243.57	1162.23 ± 245.16	<0.001
z-score BW	−0.75 ± 1.22	−0.58 ± 1.04	ns
Twins	24 (46%)	97 (52%)	ns
PN (days)	25.35 ± 10.51	20.54 ±13.55	<0.05
GA at discharge (weeks)	40.73 ± 3.24	38.96 ± 2.87	<0.001
LOS (days)	76.85 ± 30.15	67.42 ± 27.82	<0.05
Weight at discharge	2667.40 ± 572.96	2720.67 ± 582.34	ns
Weight z-score at discharge	−1.48 ± 1.07	−0.99 ± 1.01	<0.01
Mechanical ventilation (days)	7.46 ±9.27	3.41 ± 10.05	<0.01
NIV (days)	41.96 ± 32.27	31.69 ± 28.87	<0.05
ALP (UI/L)	681.48 ±246.42	492.18 ± 158.2	<0.001
P (mg/dL)	4.26 ± 1.04	6.15 ± 0.93	<0.001

**Table 2 ijms-23-10666-t002:** Comorbidities. NEC, necrotizing enterocolitis; BPD, pulmonary bronchodysplasia; IVH, intraventricular hemorrhage; ROP, retinopathy of prematurity. G1, subjects at increased risk of metabolic bone disease. G2, subjects with standard risk of metabolic bone disease.

	G1 (52)	G2 (186)	
	N(%)	N(%)	*p*
NEC	2 (3.8%)	10 (5.4%)	ns
Cholestasis	9 (17.3%)	33 (17.7%)	ns
BPD	19 (36.5%)	38 (20.4%)	0.026
IVH	5 (9.6%)	15 (8.1%)	ns
Sepsis	29 (55.8%)	89 (36.6%)	ns
ROP	4 (7.7%)	15 (8%)	ns

**Table 3 ijms-23-10666-t003:** Multivariate linear regression (R^2^ = 0.72). SD, standard deviation; BW, birth weight; PN, parenteral nutrition.

	Beta	SD	*p*
BW	0.092	0.007	0.016
PN	0.993	0.073	0.018

## Data Availability

Data available on request due to restriction eg privacy or ethical.

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
