# Peer review of "Don’t Forget the Bones: Incidence and Risk Factors of Metabolic Bone Disease in a Cohort of Preterm Infants"

_ijms, 2022, doi:10.3390/ijms231810666_

Round 1

Reviewer 1 Report

This paper is NOT suitable for publication in IJMS. Indeed, this paper is too short to catch the experimental idea. Therefore, it is extremely difficult to understand and requires much effort to understand its content. I would suggest rejection based on the suggestions below. As a recommendation, this paper can be published in other potential journals such as “Paediatrics and Child Health” after the following changes:

1- The study is quite comprehensive and has achieved very good results. Authors should write the abstract section in more detail.

2- The authors did not explain the novelty and significance of their work in the introduction part. Indeed, the introduction part is not cohesive. Topics change from sentence to sentence. The authors should follow the funnel procedure. The funnel technique for writing the introduction begins with generalities and gradually narrows your focus until you present your thesis.

3- There are some formatting mistakes in the references section, I suggest the authors to check and correct them. For example, there are incomplete references or with erroneous data, others with typos in the journal name or chemical formulae in the title.

4- The English of the whole paper is good, but some errors could still be found. Therefore, the English of the paper should be reviewed. 

Reviewer 2 Report

The aim of this study was to evaluate incidence of MBD in a cohort of VLBWI and the role of some risk factors. Good that data on risk factors/covariates for MBD were collected.

Abstract needs to include a more focused introduction/background (1-3 sentences) relevant to the goal of the paper to support the objective statement. The authors need to add sections for methodology, results, and conclusions.

Introduction-

The introduction while informative needs to be more concise and focused with what the objectives of the paper are-there are too many unrelated risk factors or background information presented.

Materials and Methods-

The time of enrollment is confusing as it says that "Preterm infants were enrolled and evaluated between the 3rd and 5th weeks of life." but in the results that 51 infants died during the first weeks of life and 238 were included in final analysis. Based on statement in the methods these would not have been considered so please update to reflect the correct methodology (1st week or 3rd week). Please confirm that no additional children died after week three (or whenever they were enrolled). Were your cutpoints for G1 and G2 based on standard clinical guidelines? Please provide a reference for that. There are too many acronyms used throughout the manuscript-I had to keep going back to the introduction to try to figure out what each of the acronyms stood for and which of the ones mentioned in the intro were used for the classification system; please reduce the number of acronyms and include in subjects paragraph what the acronyms are. Either the acronym meaning needs to be listed under the table in footnotes or just put the name in the category. Also it is unclear the metrics for classification:  "Subjects were classified as “increased risk” (G1) if P ≤ 4.5mg/dl 152 or ALP ≥ 900UI/L, or “standard risk” (G2) if P ≥ 4.5mg/dl and ALP ≤ 900UI/L." for G2 did you mean "and" as listed or "or". Also how was this tested (serum etc) and through what means-assay, etc. A lot more detail is needed in the method to standardize and for critical analysis of the data. In the table it mentions "FA" and "ALP" but is written differently in the text of the manuscript. Please correct these errors. For the statistical analysis please use the fisher's exact test; the chi-square test is not valid for cells with an N<5 subjects. 

Results-

 The study objective states" The aim of this study was to 25 evaluate incidence of MBD in a cohort of VLBWI and the role of some risk factors."  The incidence is not reported but rather a % based on a classification system. Please either report the incidence with confidence intervals or remove the word incidence from the objective statement. There are statistical tests that are reported in the results section never mentioned in the methods. Please fix this. Table 1 is not needed and could me mentioned in the results. The results read like the conclusions please fix that and just state what was found without additional discussion. For table 3 are you able to provide analysis adjusted for the characteristics in table 2? Please clarify what linear regression was done. 

Reviewer 3 Report

Abstract:

The abstract does not contain any data as presented in the results. I would recommend to present a summary of the study results in the abstract. A short description of study design, methods, patient sample, main results are essential in the abstract (in order to match to the content of the manuscript).

Patient and methods:

I miss a statement on patients with genetic abnormalities or conditions requiring surgery in the neonatal period or congenital heart disease. Were these patients excluded or included?

The groups G1 and G2 (high and low risk) were defined by a single measurement of Phosphate and Alkaline Phosphatase at age 3-5 weeks. This is a wide time range. Maybe the exact time point of measurement of these parameters can be shown in order to describe the sample cohort better and in order to show, that there was no influence on group allocation depending on the time of phosphate and ALP measurement.

Patients who died during the hospital stay were excluded from the analysis (there were 51/289 children excluded). In my opinion the outcome “death during hospital stay” is a relevant outcome – which may occur early or late during the hospital stay – the patients who die after a long course may well have a bone mineralization deficit and should be included in the analysed sample cohort in my opinion. Possibly items other than BPD may reach significance, if patients who died during the hospital stay, are not excluded.

The factor multiple gestation is not analysed in the results section. In my opinion this factor may well be relevant and deserves to be looked at.

The diagnosis retinopathy of prematurity (ROP) may be associated with MBD and should also be included in the analysis in my opinion.

I would suggest to include “multiple gestation” and ROP, and death (after week 3-5) into the analysis in table 3

The neonatal conditions “sepsis”, “NEC”, “IVH” need to be defined more precisely: Questions: Is a sepsis defined clinically or by a positive blood culture, is NEC any grade of NEC or only operatively treated NEC, IVH of any grade or higher grade (III)?

It would be interesting, to see, whether the patients had normalized parameters regarding MBD prior to discharge or in whom the parameters persisted.

Discussion:

In the discussion the prenatal aspect is mentioned. I would suggest to discuss the association between demographic risk factors and adverse outcome parameters with bone mineralization. E. g. does MBD affect respiration adversely or is BPD a factor leading to MBD? I would suggest to add a statement on recommended diagnostics and treatment in MBD, including a statement on differential diagnoses of diseases associated with MBD in children into the discussion.

Limitations:

I would suggest to include a limitations section into the manuscript (study design, sample size, heterogenic patient group).

Author contributions. This paragraph has not been dealt with, the authors contributions are not given in the text.

Formal aspects: Each abbreviation must be explained at the first use. The authors sometimes switch between present and past tense inappropriately.

Reviewer 4 Report

The authors studied the incidence and risk factors of Metabolic Bone Disease of Prematurity in 238 Very Low Birth Weight infants. The abstract does not to clearly describe the method and results. The biomarkers (phosphate, alkaline phosphatase) for diagnosing Metabolic Bone Disease of Prematurity are not comprehensive. Meanwhile, none of the bone metabolism markers, such as calcium, phosphate, PTH, and vitamin D alone, can be considered specific to Metabolic Bone Disease of Prematurity. Radiological markers and Quantitative Ultrasound are encouraged for the diagnosis. Moreover, the conclusions are not appropriately related to the results.

Round 2

Reviewer 1 Report

Accept.

Author Response

Thank you for your revision that implemented the quality of our paper.

Reviewer 2 Report

I appreciate that the authors have made detailed changes to the manuscripts based on the recommendations. I am fine with publication based on this version. Thanks!

Author Response

I wuold like to thank you for your revision that implemented the quality of our paper. 

Reviewer 3 Report

Please check: The sentence line 142 ff is unclear. I think it intends to express: Indeed, wheras no patients with birth weight >1000g developed the overt disease, ....

Author Response

Thank you for your suggestion. We have modified the sentences according to your review. 

Reviewer 4 Report

 The abstract should be rewritten following the style of the journal.

 The result of multivariate linear regression analysis may be shown to the readers. 

For the description: Five patients (2%) presented radiological signs of osteopenia. It should be essential to show the pictures of the five radiological signs of the infants in the result section.

The relationship between the biomarkers, radiological markers, and Quantitative Ultrasound is encouraged to be mentioned in the introduction section and be discussed in the discussion section. 

The style of the language is encouraged to improve for publication.

 For the conclusions, I would suggest as following:

Despite nutritional approach and early supplementation with calcium, phosphorus and vitamin D, preterm infants have an increased risk of MBD, especially if born with a weight below 1000 g. Screening for MBD based on serum ALP could be helpful in clinical practice to identify subject at increased risk. Further studies are needed to explore the pathophysiological mechanisms underlying the development of MBD, in order to implement preventive therapies to reduce the incidence of this complication in a susceptible population which is still elevated.

Author Response

Thank you for your revision. We have modified the paper according to your suggestion as you can read in the attachment.

Thank you for you suggestion that implemented the quality of our paper 

Round 3

Reviewer 4 Report

There have been significant improvements in the manuscript, and I would suggest its publication!